# Shock Index Predicts Outcome in Patients with Suspected Sepsis or Community-Acquired Pneumonia: A Systematic Review

**DOI:** 10.3390/jcm8081144

**Published:** 2019-07-31

**Authors:** David J. Middleton, Toby O. Smith, Rachel Bedford, Mark Neilly, Phyo Kyaw Myint

**Affiliations:** 1Institute of Applied Health Sciences, University of Aberdeen, Scotland AB25 2ZD, UK; 2Nuffield Department of Orthopaedics, Rheumatology and Musculoskeletal Sciences, University of Oxford, Oxford OX3 7LD, UK

**Keywords:** shock index, sepsis, pneumonia, prognosis

## Abstract

Background: To improve outcomes for patients who present to hospital with suspected sepsis, it is necessary to accurately identify those at high risk of adverse outcomes as early and swiftly as possible. To assess the prognostic accuracy of shock index (heart rate divided by systolic blood pressure) and its modifications in patients with sepsis or community-acquired pneumonia. Methods: An electronic search of MEDLINE, EMBASE, Allie and Complementary Medicine Database (AMED), Cumulative Index to Nursing and Allied Health Literature (CINAHL), Open Grey, ClinicalTrials.gov and the WHO International Clinical Trials Registry Platform (WHO ITRP) was conducted from conception to 26th March 2019. Eligible studies were required to assess the prognostic accuracy of shock index or its modifications for outcomes of death or requirement for organ support either in sepsis or pneumonia. The methodological appraisal was carried out using the Downs and Black checklist. Evidence was synthesised using a narrative approach due to heterogeneity. Results: Of 759 records screened, 15 studies (8697 patients) were included in this review. Shock index ≥ 1 at time of hospital presentation was a moderately accurate predictor of mortality in patients with sepsis or community-acquired pneumonia, with high specificity and low sensitivity. Only one study reported outcomes related to organ support. Conclusions: Elevated shock index at time of hospital presentation predicts mortality in sepsis with high specificity. Shock index may offer benefits over existing sepsis scoring systems due to its simplicity.

## 1. Introduction

Sepsis is defined as life-threatening organ dysfunction caused by a dysregulated host response to infection [1]. The incidence of sepsis and septic shock are rising and, despite considerable advances in understanding, morbidity and mortality remain high [2]. Even in high income countries, the mortality rate of hospital-treated sepsis exceeds 20%, reflecting a global burden of more than 5.3 million deaths annually [3]. Community-acquired pneumonia (CAP) is the most common source of sepsis, accounting for over 40% of cases in major studies, and is responsible for disproportionate morbidity and mortality [4]. Early protocolised resuscitation of patients with sepsis is known to improve outcomes [5,6], whilst prognosis is worse in those in whom identification of critical illness is delayed [7,8,9].

Therefore, the early identification of patients with sepsis who are at high risk of deterioration or death is vital to enable appropriate initial management and consideration of escalation to higher level care. Widely used sepsis screening scores, such as the Systemic Inflammatory Response Syndrome (SIRS) criteria and the National Early Warning Score (NEWS), are of limited prognostic utility due to inadequate specificity [10,11]. The recently recommended quick Sequential Organ Failure Assessment (qSOFA) score [1] is intended to be used as a severity assessment tool but fails to identify half of patients with sepsis who will subsequently die [10,11,12]. The calculation of these scores requires training and can be relatively time consuming.

The shock index (SI), derived using two simple physiological measures, heart rate divided by systolic blood pressure, is a simple bedside assessment of cardiovascular status which has been used to predict adverse outcomes in patients with haemorrhagic shock [13,14] and pulmonary embolism [15]. The normal range is 0.5–0.7 [16]. Various modifications of the shock index have been proposed to improve accuracy, including the Adjusted Shock Index (ASI; SI adjusted for body temperature) [17] and the Modified Shock Index (MSI; heart rate/mean arterial pressure) [18]. These have been applied to patients with CAP, and existing CAP severity scores modified to incorporate shock index [16,17,19,20,21]. The usefulness of shock index or its modifications for predicting outcomes in sepsis and CAP has been previously investigated, but no systematic review has synthesised the evidence to determine the prognostic utility of shock index in sepsis and CAP. 

The aim of this systematic review was therefore to examine the current evidence-base and determine whether shock index or its modifications are useful predictors of morbidity and mortality in hospitalised adult patients with: (i) suspected sepsis; or (ii) community-acquired pneumonia. 

## 2. Materials and Methods

The systematic review and meta-analysis were undertaken in accordance with the Preferred Reporting Items for Systematic Reviews and Meta-Analyses (PRISMA) recommendations. The protocol was registered through the PROSPERO database (registration number: CRD42018096473).

### 2.1. Selection Criteria

The eligibility criteria are presented below: 

Inclusion: (i) Study participants comprise adult (≥18 years) patients who were hospitalised with a diagnosis of sepsis or CAP. (ii) Measurements reported include shock index, adjusted shock index or modified shock index. (iii) Outcome measures reported include mortality, requirement for Intensive Care admission, requirement for vasoactive support, renal replacement therapy or mechanical ventilation. (iv) Prospective or retrospective cohort study design.

Exclusion: Lack of available full-text report.

### 2.2. Search Strategy

The databases MEDLINE, EMBASE, Allie and Complementary Medicine Database (AMED), Cumulative Index to Nursing and Allied Health Literature (CINAHL), Open Grey, ClinicalTrials.gov and the WHO International Clinical Trials Registry Platform (WHO ITRP) were searched from their inception to 21st November 2016. This was updated on 26th March 2019. The search strategy is presented in Appendix A. A hand search of the reference list of all relevant reviews and primary articles was performed to identify any articles not captured by the electronic searches. Restrictions were not applied, such as age of publication, language or number of included patients. Only full-text reports were considered. For relevant conference abstracts meeting eligibility criteria, authors were contacted to determine the existence of any related full-text report. 

### 2.3. Study Selection 

All search results were independently screened by a minimum of two reviewers (DM, RB, MN). The full texts of those considered potentially eligible were obtained and reviewed against eligibility criteria by the same individuals. Any disagreement on study eligibility was resolved through discussion with senior review team members (TOS, PKM). 

### 2.4. Data Extraction 

All data were independently extracted by two reviewers (DM, RB) using a piloted data extraction template, with disagreements resolved by discussion. Data were extracted for all studies included the following: year of publication, study design, setting, inclusion/exclusion criteria, sample size, age and sex of participants, source of sepsis, measurement of interest, threshold values for binary classification of cohort, outcome measures used, percentage mortality of cohort, test characteristics for prediction of outcomes of interest (where data allowed).

### 2.5. Quality Assessment 

Risk of bias and study quality was assessed using the Downs and Black tool for non-randomised controlled trials [22], which was applied to eligible studies by two independent reviewers (DM, RB). Disagreements were resolved through discussion. Questions 8, 14, 19, 23 and 24 of the Downs and Black tool are not relevant to cohort studies, so were omitted. 

### 2.6. Data Analysis

Study heterogeneity was determined by visual inspection of the data extraction table by two reviewers (DM, RB), assessing for between-study variability/similarity, study design, participant characteristics, reported measurements and outcomes. This identified substantial inter-study heterogeneity for all outcomes. Consequently, data were analysed using a narrative approach. 

## 3. Results

### 3.1. Search Results

The PRISMA flow diagram summarising the search results is presented as Figure 1. A total of 754 citations were identified. From these, 50 were potentially eligible. Based on a full-text review of these studies, 15 satisfied the pre-defined eligibility criteria and were included. 

### 3.2. Quality Assessment 

A summary of the Downs and Black [22] quality assessment is presented in Appendix A. There was marked variability in the quality of the evidence. Papers frequently successfully reported the study objectives (100%), measurements of interest (100%), main outcome measures (93%), population characteristics (87%) and probability values of their inferential analysis (93%). However, papers’ main findings were less well described (60%), and few recorded (33%) or adjusted for potential confounders (27%). External validity of the evidence was generally low, with few studies conducted in large diverse populations (20%). Risk of bias in the evidence was low with the vast majority studies recruiting all patients from the same population at the same time point (93%), and with outcomes assessed at a standardised time-point (87%). Sixty percent of studies reported a sample size calculation. 

### 3.3. Characteristics of Studies Included

A summary of the characteristics of included studies is presented in Table 1.

#### 3.3.1. Sepsis

Nine studies (*n* = 7759) investigated patients with sepsis. Of these, seven were retrospective and eight were of single-centre design. Seven assessed an Emergency Department (ED) population, with remaining studies using pre-hospital [23] and medical ICU [24] cohorts. There was marked inter-study heterogeneity in criteria used to identify cohorts and in disease severity, with reported mortality rates ranging from 5 to 54%. Seven studies included patients with sepsis of any source, whilst one investigated a specific cohort of elderly patients with influenza [25], and another only included patients with septic incomplete miscarriage [26].

Eight studies (*n* = 7181) assessed the prognostic utility of shock index, though there was considerable variation in the threshold values used, ranging from ≥ 0.7 to ≥ 1.0. Five of these studies measured shock index for a single time point (on admission) and others considered serial ED measurements [27] or two interval measurements [26,28]. Outcomes of interest included mortality (100%), intensive care admission (11%) [23] and requirement for vasoactive support (11%) [27]. 

#### 3.3.2. Community-Acquired Pneumonia

Six studies (*n* = 938) investigated patients with CAP. Notably, three of these studies (by one of the co-authors of this work) utilised the same population of 190 patients, variously applying SI and ASI [29], the CURSI and CURASI scores (modifications of the CURB-65 score where the blood pressure element replaced by SI or ASI and age is omitted) [30] or the CARSI and CARASI scores (modifications of the CURB65 score where the blood pressure element is replaced by SI or ASI and urea is omitted) [17] to patients with CAP. Two further studies [19,21] assessed the prognostic utility of CURSI and CURASI, whilst one [20] compared the SIPF score (comprising SI and PaO2/FiO2 ratio) to CURB-65 and the Pneumonia Severity Index (PSI). The diagnostic criteria for CAP were homogenous across studies but there was variation in disease severity, with rates of mortality ranging from 8 to 28%. 

### 3.4. Shock Index in Patients with Sepsis

#### 3.4.1. Shock Index as a Predictor of Mortality in Patients with Sepsis

Results are shown in Table 2. In four of five adequately powered studies [25,31,32,33] where shock index was measured at the time of ED admission, there was a positive correlation between elevated shock index and mortality. The largest studies, both including ED patients who had blood cultures taken, reported odds ratios for mortality of 2.0 (1.8–2.9) [31] and 3.0 (1.8–4.2) [33] using a threshold of SI ≥ 1.0. Other studies reported that serial measurements of shock index [27] or two interval measurements [26,28] allowed more accurate mortality prediction than a single measurement, though some of these studies were methodologically poor. Only one study [31] reported outcomes for shock index using two different threshold values (≥ 0.7 and ≥ 1.0) with the higher cut-off predictably increasing specificity at the expense of sensitivity. Jamies et al. [32] did not report specific outcome data for shock index, but included SI ≥ 1.5 in a multivariate model that predicted 28-day mortality with moderate accuracy (AUROC 0.75) in a population of ED patients. 

#### 3.4.2. Shock Index as a Predictor of Morbidity in Sepsis

Two studies investigated the use of shock index as a predictor of morbidity in sepsis. Both reported an association between elevated shock index and increased morbidity, with one [23] finding that a pre-hospital shock index ≥ 0.7 was a strong predictor of ICU admission (OR 5.96; 95% CI1: 49–25.78). Wira et al. [27] reported that sustained shock index elevation (≥ 0.8 for > 80% of ED measurements) was a predictor of vasopressor dependence within 72 h of admission, compared to patients with non-sustained elevation in shock index (OR 4.42; 95% CI: 2.28–8.55). No studies reported on the association between shock index and requirement for renal replacement therapy or mechanical ventilation.

#### 3.4.3. Modifications of Shock Index in Patients with Sepsis

One study reported a weak association between a sustained but not isolated elevation in modified shock index ≥ 1.3 in medical ICU patients during the first 6 h of admission (OR 1.13; 95% CI: 1.02–1.26) [24].

### 3.5. Shock Index in Patients with CAP

#### 3.5.1. Shock Index and Adjusted Shock Index as a Predictor of Mortality and Morbidity in Patients with CAP

Two studies reported on the prognostic utility of shock index for mortality in patients with CAP [21,29]. SI ≥ 1 predicted mortality with a low sensitivity and relatively high specificity, similar to its performance in the patients with undifferentiated sepsis (Table 3). ASI ≥ 1.0 did not perform significantly better in the only cohort that reported relevant data [29], though the study may not have been sufficiently powered to show this. No studies reported on the association between shock index or adjusted shock index and morbidity in patients with CAP.

#### 3.5.2. Modifications of Shock Index and Adjusted Shock Index in Patients with CAP

Three studies reported on the performance of the CURSI score for predicting mortality in patients with CAP (Appendix A) [19,21,30]. The CURSI score did not outperform CURB65 in any study, and in the largest cohort [21] its sensitivity was inferior to that of CURB65. In two studies [19,30], the CURSI and CURASI scores performed equivalently. Using the same patients population as Myint et al. [30], Musonda et al. [17] found that the CARSI and CARASI scores performed equivalently with CURB65, although with a non-significant trend towards lower sensitivity. One poor quality study [20] reported that the SPIF score performed equivalently to CURB65 and the Pneumonia Severity Index for prediction of mortality or ICU admission in patients with CAP. 

## 4. Discussion

The early assessment of disease severity is vital in patients with suspected sepsis and CAP, due to the high rate of mortality associated with these conditions, and the potential modifiability of the disease process and outcome [5,6]. The early identification of patients at high risk of mortality enables the appropriate allocation of scarce recourses, such as intensive care beds. The findings of this review suggest that initial elevated shock index is a moderately accurate predictor of mortality in adult patients with suspected sepsis. The two large studies found a similar effect size, reporting mortality odds ratios of 2.24 (1.81–2.91) [31] and 2.80 (1.80–4.20) [33] using a threshold of SI ≥ 1.0. Whilst smaller studies [23,26,27] did not detect a significant association between initial shock index and mortality, they are at high risk of type 2 error. Elevated initial shock index was more strongly predictive of mortality in a cohort of elderly Taiwanese patients with influenza and a low case mortality rate [25], though the generalisability of this study is low given the highly specific population investigated. In patients with CAP, the most common cause of sepsis, elevated shock index appears to predict mortality with similar accuracy to that seen in patients with sepsis [21,29]. 

There was broad agreement across studies that SI ≥ 1.0 has low sensitivity and relatively high specificity for mortality prediction, suggesting that it may be useful as a “rule in” test when positive to trigger prompt escalation of care, whilst a negative result should not be considered reassuring. However, it is unclear whether there is a practical application for shock index in sepsis prognostication given that NEWS, SIRS and qSOFA are all likely to have superior sensitivity [11,12], and qSOFA has a comparable specificity [12]. Shock Index may identify some high-risk patients missed by qSOFA, as this score integrates fewer haemodynamic variables than the others (blood pressure only).

There was weak evidence that sustained elevation of SI following initial resuscitation may be more predictive of adverse outcomes [26,27,28], though this requires validation in methodologically rigorous and adequately powered studies. Furthermore, the requirement for periodic measurement of SI may not be practical in the ED where resource is limited and early decision making on patient disposition is vital to enable prompt transfer. 

Modified shock index is thought to be more accurately predictive of mortality than shock index in undifferentiated Emergency Department patients [18], but it has only been assessed in sepsis patients in ICU, where sustained elevation was weakly predictive of mortality [24]. The prognostic utility of adjusted shock index has only been reported in one study in patients with CAP [29], where it performed equivalently with SI for prediction of mortality. The assessment of these indices in larger populations of sepsis patients is required. Attempts to integrate SI and ASI into more complex clinical severity scores for CAP [20,21,30] has not yet demonstrated convincing benefits over the CURB-65 score, though the CARSI score may offer an equivalent performance without the requirement for urea testing [17], and thus can be useful in primary care setting to assess CAP severity.

In studies that adjusted for covariates (e.g., age, sex, physiological observations [29,32,33]), shock index was consistently found to be an independent predictor of mortality. This is also seen in conditions as diverse as stroke [16], myocardial infarction [34], pulmonary embolism [15], and haemorrhagic shock [13,14], highlighting the common importance of haemodynamic instability as an antecedent mortality. Unselected patients who present to hospital with overt shock (with hypotension) have a much higher mortality than those who present with covert shock (without hypotension) [35]. The value of SI lies in its ability to identify the cohort of patients with covert shock at an earlier stage in their disease process, prior to the failure of their innate compensatory mechanisms, thus maximising the opportunity for meaningful intervention.

The strengths of this systematic review include conducting the comprehensive literature search utilising several databases, and the duplication of literature screening, selection and data extraction by two reviewers to ensure accuracy. Notable limitations include the predominantly retrospective evidence base, and marked heterogeneity in study populations, index measurements and reporting of outcomes. This precluded a meaningful estimation of pooled effects size by meta-analysis, or assessment for publication bias by funnel plot. The variability of study populations and mortality rate, and the relative lack of multi-centre studies may limit the generalisability of our findings. There was also little or no data for some of the pre-specified morbidity outcomes, and there was insufficient data to draw conclusions on the utility of ASI and MSI in our populations of interest.

## 5. Conclusions

In conclusion, shock index is a moderately accurate predictor of mortality in adult patients with sepsis and CAP, which may have utility as a “rule-in” tool to identify high-risk patients. The simplicity and rapidity with which it can be calculated is an advantage over existing sepsis scoring systems, particularly in resource-limited settings. Combining shock index with other established and rapidly available predictors of prognosis, such as lactate and neutrophil/lymphocyte ratio, may further improve prognostic accuracy. Future research should focus on: (i) whether the integration of SI into current decision-making tools for sepsis or CAP can augment their accuracy; and (ii) the use of the electronic recoding of vital signs to determine whether trends in shock index better predict outcomes than one-off measurements.

## Figures and Tables

**Figure 1 jcm-08-01144-f001:**
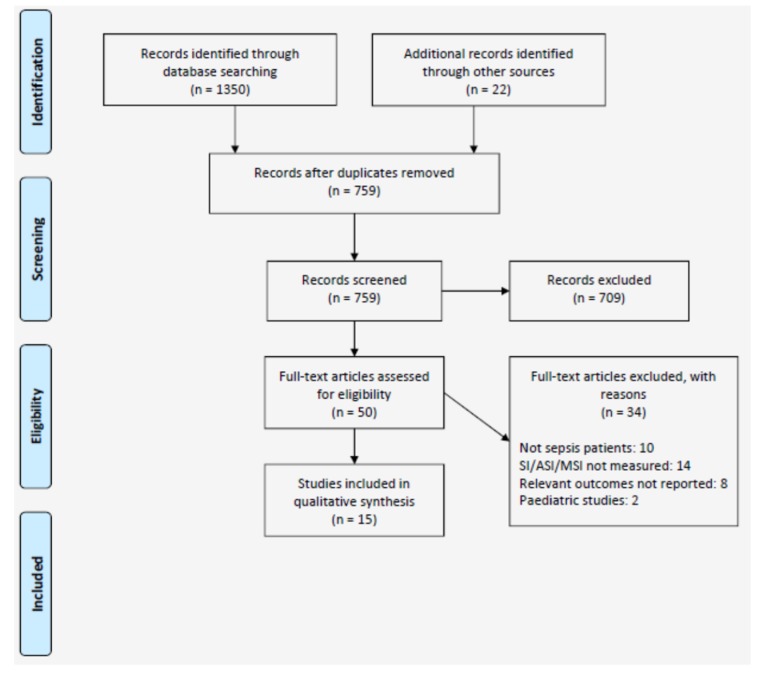
Preferred Reporting Items for Systematic Reviews and Meta-Analyses (PRISMA) flow diagram.

**Table 1 jcm-08-01144-t001:** Characteristic of included studies.

Author/Year	Design	*n*	Location	Setting	Study Population	Mortality (%)	Admitted to ICU (%)	Index Test and Range	Outcome(s) of Interest
**Sepsis studies**									
**Baez et al., 2013 [23]**	Retrospective cohort	63	USA; 1 centre	Pre-hospital	Adults (≥ 18) with ICD-9 diagnostic code of sepsis, severe sepsis or septic shock	34.9	68.3	SI ≥ 0.7	In-hospital mortality ICU admission
**Berger et al., 2013 [31]**	Retrospective cohort	2524	USA; 1 centre	ED	Adults (≥ 21) screened for sepsis using standardised blood order	13.5	-	SI ≥ 0.7; SI ≥ 1	28-day mortality
**Chung et al., 2019 [25]**	Retrospective cohort	409	Taiwan; 1 centre	ED	Elderly patients (≥ 65) with pyrexia confirmed influenza	4.9	-	SI ≥ 1.0	30-day mortality
**Jaimes et al., 2005 [32]**	Prospective cohort	533	Colombia; 2 centres	ED	Adults (≥ 16) suspected sepsis (2012 definition)	18.9	14.1	SI; full range	Multivariable modelling to predict 28-day mortality
**Jayarakash et al., 2018 [24]**	Retrospective cohort	578	USA; 1 centre	Medical ICU	Adults (≥ 18) admitted to ICU with severe sepsis or septic shock (2012 definition)	19.9	100	MSI ≥ 1.3	In-hospital mortality
**Lombaard et al., 2015 [26]**	Audit	47	South Africa; 1 centre	Maternity ward	Adult patients with septic incomplete abortion	19.2	-	SI ≥ 1.0	In-hospital mortality
**Talmor et al., 2007 [33]**	Prospective cohort	3260	Israel; 1 centre	ED	Adults (≥ 18) who had blood cultures taken	4.7	12	SI ≥ 1.0	In-hospital mortality or ICU
**Wira et al., 2014 [27]**	Retrospective cohort	295	USA; 1 centre	ED	Adults (≥ 18) with severe sepsis (2012 definition)	15.6	-	SI ≥ 0.8 for ≥ 80% of ED values	Vasopressor dependence by 72-h 28-day mortality
**Yussof et al., 2012 [28]**	Retrospective cohort	50	Malaysia; 1 centre	ED	Adults (≥ 18) triaged to resuscitation area with sepsis or septic shock (2012 definition)	54%	-	SI; entire range at presentation and 2 h	In-hospital mortality
**Community-acquired pneumonia (CAP) studies**									
**Curtain et al., 2013 [19]**	Prospective cohort	95	UK; 1 centre	Hospital ward	Adults (≥ 18) admitted with CAP (symptoms and new CXR shadow)	8.4%	9.5%	SI and ASI ≥ 1.0, as part of CURSI, CURASI score	6-week mortality
**Eldaboosy et al., 2015 [20]**	Retrospective cohort	100	Egypt and Saudi Arabia; 2 centres	Hospital ward	Adults admitted with CAP (symptoms and new CXR shadow)	9%	34%	SI ≥ 0.7, as part of SIPF score	In-hospital mortality ICU admission
**Musonda et al., 2011 [17]**	Prospective cohort	190	UK; 3 centres	AMAU	Adults (≥ 18) admitted with CAP (symptoms and new CXR shadow)	28.4%	-	SI ≥ 1.0, as part of CARSI and CARASI score	42-day mortality
**Myint et al., 2010 [30]**	Prospective cohort	190	UK; 3 centres	AMAU	Adults (≥ 18) admitted with CAP (symptoms and new CXR shadow)	28.4%	-	SI ≥ 1.0, as part of CURSI and CURASI score	42-day mortality
**Nullmann et al., 2014 [21]**	Retrospective cohort	553	Germany; 1 centre	Hospital ward	Adults (≥ 18) admitted with CAP (symptoms and new CXR shadow)	10.7%	10.5%	SI ≥ 1.0, as part of CURSI score	30-day mortality
**Sankaran 2011 [29]**	Prospective cohort	190	UK; 3 centres	AMAU	Adults (≥ 18) admitted with CAP (symptoms and new CXR shadow)	28.4%	-	SI ≥ 1.0 ASI ≥ 1.0	42-day mortality

**Table 2 jcm-08-01144-t002:** Association between SI and mortality in patients with sepsis.

Author/Year	*n*	SI Threshold	Mortality (%)	Test Characteristics for Prediction of Mortality				
				Sens	Spec	PPV	NPV	OR
**Baez et al., 2013 [23]**	63	SI ≥ 0.7	34.9	-	-	-	-	1.66 (0.59–4.65)
**Berger et al., 2013 [31]**	2524	SI ≥ 0.7	13.5	0.71 (0.66–0.76)	0.41 (0.39–0.43)	0.17 (0.16–0.18)	0.89 (0.88–0.91)	1.68 (1.32–2.14)
	-	SI ≥ 1.0	-	0.32 (0.27–0.36)	0.79 (0.77–0.81)	0.23 (0.20–0.26)	0.85 (0.84–0.86)	2.24 (1.81–2.91))
**Chung et al., 2019 [25]**	409	SI ≥ 1.0	4.9	0.30 (0.12–0.54)	0.94 (0.91–0.96)	0.21 (0.11–0.36)	0.96 (0.95–0.97)	6.78 (2.39–19.29)
**Lombbard et al., 2015 [26]**	47	SI ≥ 1.0	19.2	0.77 (0.40–0.97)	0.29 (0.15–0.46)	0.21 (0.15–0.28)	0.85 (0.25–0.54)	1.43 (0.26–7.97)
**Talmor et al., 2007 [33]**	3260	SI ≥ 1.0	4.7	-	-	-	-	2.8 (1.8–4.2)
**Wira et al., 2014 [27]**	295	SI ≥ 0.8 for > 80% of ED measurements	15.6	0.59 (0.43–0.73)	0.55 (0.48–0.61)	0.19 (0.15–0.24)	0.88 (0.83–0.91)	1.71 (0.90–3.23)
**Yussof et al., 2012 [28]**	50	SI ≥ 1.2 on admission	54	0.73	0.45	-	-	-
	-	SI ≥ 1.0 at 2 h	-	0.81	0.79	-	-	-

**Table 3 jcm-08-01144-t003:** Association between SI/ASI and mortality in patients with CAP.

Author/Year	*n*	SI Threshold	Mortality (%)	Test Characteristics for Prediction of Mortality				
				Sensitivity	Specificity	PPV	NPV	OR
**Nullmann et al., 2014 [21]**	**443**	**SI ≥ 1.0**	10.7	0.25 (0.15–0.38)	0.87 (0.84–0.90)	0.18 (0.12–0.26)	0.92 (0.90–0.92)	2.72 (1.42–5.21)
**Sankaran 2011 [29]**	190	SI ≥ 1.0	28.4	0.28 (0.16–0.41)	0.83 (0.75–0.89)	0.39 (0.27–0.54)	0.74 (0.71–0.78)	1.89 (0.90–3.98)
	-	ASI ≥ 1.0	-	0.22 (0.12–0.36)	0.90 (0.83–0.94)	0.45 (0.30–0.63)	0.74 (0.71–0.77)	2.49 (1.07–5.81)

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
