# Peer review of "Shock Index Predicts Outcome in Patients with Suspected Sepsis or Community-Acquired Pneumonia: A Systematic Review"

_jcm, 2019, doi:10.3390/jcm8081144_

Round 1

Reviewer 1 Report

This is a well written paper. The authors demonstrate that the SI has high specificity but lower sensitivity for predicting mortality in patients with sepsis. It is therefore useful as a "rule in" but not "rule out" test. In the conclusion I would add that the predictive value of the SI may be improved by the addition of other available parameters readily available soon after presentation; ie. lactate, or neutrophil/lymphocyte ratio etc.Additional, studies would be required to support his contention.     

Author Response

Dear Reviewer, 

Thank you for your timely feedback. I have amended lines 256-258 of the conclusion as you recommended. 

Best wishes, 

David 

Reviewer 2 Report

The authors reviewed the value of the shock index in predicting mortality in sepsis and CAP. As expected there was a good predictive value.

The data are well      expected. Since both heart rate and systolic blood pressure are correlated      with mortality, their ratio should do it better than one of these two abnormalities      alone. One can realize that a patient with systolic blood pressure of 60      mmHg and a heart rate of 140/min has a high risk of mortality. The      question is why is it important to predict mortality?

A second figure should      be included representing mortality vs. shock index in the various studies.

Why sepsis and CAP are      combined in the study is unclear to me – why not VAP, peritonitis, UTI...and      other infections?

Why this manuscript should      be evaluated through a fast track is unclear.

Author Response

Dear Reviewer,

Thank you for your timely and helpful feedback. In reply to your comments:

1.      Whilst I agree it is not surprising that elevated shock index is associated with mortality with patients with sepsis and community acquired pneumonia, it is important to quantify the strength of this association in order to allow for appropriate weight to be given to this finding by treating clinicians.  For example, a variable that was associated with a mortality odds ratio of 20 would be given much greater prominence in risk-assessment than a variable with a mortality odds ratio of 2.  Mortality is an important patient centred outcome, and is a clear surrogate marker of disease severity.  Thus, early prediction of high mortality risk allows the initiation of early interventions aimed at preventing this, such as ITU observation. I have amended lines 199-202 of the discussion to try and emphasise these points.

2.      I agree a graphical representation of mortality vs shock index per study would be informative. Unfortunately, only 2 studies (Chung et al and Talmor et al) report the mean SI for their overall cohort, with most studies splitting their report of population characteristics by shock index (for example above or below 1), or by outcome (e.g. mortality or not).  Therefore, it is not possible to produce such a figure.

3.      CAP was included as a separate condition of interest in this study because it is the most common cause of sepsis, and is responsible for disproportionate morbidity and mortality compared to other sources of sepsis. I have edited lines 35-36 of the introduction and line 201 of the discussion to try and emphasise this point.  

Best wishes,

David